# Effects of Protein Supplementation Combined with Exercise Training on Muscle Mass and Function in Older Adults with Lower-Extremity Osteoarthritis: A Systematic Review and Meta-Analysis of Randomized Trials

**DOI:** 10.3390/nu12082422

**Published:** 2020-08-12

**Authors:** Chun-De Liao, Yen-Tzu Wu, Jau-Yih Tsauo, Pey-Rong Chen, Yu-Kang Tu, Hung-Chou Chen, Tsan-Hon Liou

**Affiliations:** 1School and Graduate Institute of Physical Therapy, College of Medicine, National Taiwan University, Taipei 100025, Taiwan; 08415@s.tmu.edu.tw (C.-D.L.); yenwu@ntu.edu.tw (Y.-T.W.); jytsauo@ntu.edu.tw (J.-Y.T.); 2Department of Physical Medicine and Rehabilitation, Shuang Ho Hospital, Taipei Medical University, New Taipei City 235041, Taiwan; 10462@s.tmu.edu.tw; 3Physical Therapy Center, National Taiwan University Hospital, Taipei 100229, Taiwan; 4Department of Dietetics, National Taiwan University Hospital, Taipei 100225, Taiwan; prchen@ntuh.gov.tw; 5School of Nutrition and Health Sciences, Taipei Medical University, Taipei 110301, Taiwan; 6Institute of Epidemiology and Preventive Medicine, College of Public Health, National Taiwan University, Taipei 100025, Taiwan; yukangtu@ntu.edu.tw; 7Department of Physical Medicine and Rehabilitation, School of Medicine, College of Medicine, Taipei Medical University, Taipei 110301, Taiwan

**Keywords:** osteoarthritis, sarcopenia, arthroplasty, protein supplement, exercise training, muscle mass, physical function

## Abstract

Aging and osteoarthritis (OA) are associated with a high risk of muscle mass loss, which can lead to physical disability. This study investigated the effectiveness of protein supplementation combined with exercise training (PS + ET) in improving muscle mass and functional outcomes in older adults with lower-limb OA. A comprehensive search of online databases was performed to identify randomized controlled trials (RCTs) on the effectiveness of PS + ET in older adults with hip or knee OA. Meta-analysis and risk of bias assessment of the included RCTs were conducted. Six RCTs were included in this systemic review; they had a median (range/total) Physiotherapy Evidence Database (PEDro) score of 7 (6–9) out of 10, respectively. Five RCTs that enrolled patients who underwent total joint replacement were included in this meta-analysis. The PS + ET group exhibited significant improvements in muscle mass (standard mean difference [SMD] = 1.13, *p* < 0.00001), pain (SMD = 1.36, *p* < 0.00001), and muscle strength (SMD = 0.44, *p* = 0.04). Our findings suggest that PS + ET improves muscle mass, muscle strength, and functional outcomes and reduces pain in older adults with lower-limb OA, particularly in those who have undergone total joint replacement.

## 1. Introduction

Osteoarthritis (OA), especially in joints in the lower extremity, is one of the most prevalent musculoskeletal diseases in older adults [1]. Its prevalence increases rapidly from the sixth to ninth decades of life, and it is a major health issue at individuals and population levels [2]. In particular, hip or knee OA is associated with impaired functional activity, potentially leading to disability [3].

Deficits in muscle volume and function have been observed in older adults with mild to moderate hip OA [4] as well as in people with mild to severe knee OA [5]. Such deterioration of muscle mass occurring with disease progression has been attributed to sarcopenia, a condition associated with muscle attenuation in older adults [6]. Recently, it has been reported that older adults with knee OA are at a high risk of sarcopenia [7]. This finding of high risk is based on previous observations that older adults with OA have lower appendicular lean mass in the lower limbs relative to healthy controls [8], and that low skeletal muscle mass is independently associated with radiographic knee OA [9]. Because lower-limb lean mass is closely associated with muscle power in OA [10] and because low skeletal muscle mass is closely associated with physical difficulty and poor health status among older adults [11,12], sarcopenia may lead to physical decline through muscle weakness during OA progression. Therefore, the maintenance of muscle strength and the prevention of sarcopenia are crucial for enabling older adults with hip or knee OA to successfully perform physical tasks.

OA has been recognized as a major musculoskeletal disease [13]. The management of mild to moderate OA comprises multidisciplinary interventions, including pain medications and nonpharmacological treatments; however, for end-stage OA, total joint replacement is recommended [14]. Several recent studies have indicated physical activity and nutrition as nonpharmacological and preventive treatments for OA and sarcopenia [15,16,17,18,19]. Among treatment interventions for OA, exercise training (ET) has been recommended as the first-line treatment [20]. In addition, conservative ET—administered in combination with a variety of training tools, such as kinesio tape [21] and whole-body vibration (or electromyostimulation) [19]—has been effectively employed to improve muscle mass, muscle strength, and physical function [19,22]. Additionally, dietary interventions, such as dietary protein or protein supplementation (PS), have been incorporated into the multidisciplinary management of OA [23,24,25]. Studies have noted that 30.3%–65.1% of older adults with knee or hip OA had a daily protein intake lower than the recommended daily allowance of 0.8 g/kg/day [26,27]. PS and protein-based diet interventions are believed to additionally increase the efficacy of ET in older adults [28,29]. However, inconsistent results have been obtained regarding the effectiveness of PS combined with ET (PS + ET), specifically in older adults with OA or in individuals who recently underwent total joint replacement. Previous studies have shown that PS + ET significantly improved muscle mass [30] and strength [31,32] in older adults with OA, but other studies have not [33,34]. Because older adults with OA have a high risk of sarcopenia [7], which may further affect postoperative outcomes in those who have recently undergone total joint replacement [33,34,35], the skeletal muscle plays an important role in functional recovery after total joint replacement and has thus been targeted in the management of OA [7,36,37]. Therefore, determining the effectiveness of PS + ET in preserving muscle mass in older adults with OA is crucial, especially in those who have recently undergone total joint replacement.

Evidence regarding the effects of PS + ET on healthy, sarcopenic, and frail older adults has been well established by previous systematic reviews and meta-analyses [38,39,40]; however, few studies have focused on older adults with OA or those who underwent total joint replacement. Thus, this study examined the effects of PS + ET on muscle mass and functional outcomes in older adults with OA in the lower extremity.

## 2. Method

### 2.1. Design

The present study was conducted in accordance with guidelines of the Preferred Reporting Items for Systematic Reviews and Meta-Analysis (PRISMA) [41]. The study protocol has been registered at PROSPERO (registration number: CRD42020176748). In this study, comprehensive searches of online databases were conducted, namely PubMed, EMBASE, the Cochrane Library Database, the Physiotherapy Evidence Database (PEDro), China Knowledge Resource Integrated Database, and Google Scholar. The secondary sources referred to in this study comprised papers cited by the articles obtained from the database searches. No limitations were imposed on publication year and language to minimize publication and language biases. Two authors (CDL and HCC) independently searched for relevant articles, screened them, and extracted data. Any disagreement between the authors was resolved through a consensus reached with the other team members (YTW and THL), who acted as arbitrators.

### 2.2. Search Strategy

The following keywords were used to identify participant characteristics: (“older adults” or “elder adults”) and (“osteoarthritis” or “arthroplasty”). The following keywords were used to identify the study intervention: (“exercise training” or “physical activity”) and (“protein/amino-acid/nutrient supplementation” or “nutrition intervention”). The search formulas for each database are detailed in Appendix A.

### 2.3. Selection Criteria of Studies

Trials were included if they met all of the following criteria: (1) the study was a randomized controlled trial (RCT); (2) the study enrolled participants who were aged ≥50 years, had a radiographic diagnosis of hip or knee OA with a Kellgren–Lawrence (K-L) grade of ≥II for disease severity [42], and had undergone total joint replacement (or not); (3) the experimental groups received PS + ET; (4) the exercise intervention included either resistance-based ET (RET) alone or a multicomponent exercise regime (MET) that comprised RET, aerobic exercise, balance training, and physical activity training; (5) the control group received a comparative intervention, including a placebo supplement, PS alone, ET alone, or none of the above (i.e., regular care); (6) the supplementation intervention involved protein sources, including whey protein, milk protein, leucine, and casein, whether consumed alone or in combination with other nutrients (creatine and amino acids); (7) the PS + ET intervention was applied either preoperatively or postoperatively to participants who underwent arthroplasty or replacement surgery for the knee or hip joint; and (8) the study reported measures on at least one of the primary outcomes of muscle mass, pain, and muscle strength. The definitions of these outcomes are provided in the following subsections.

Studies were excluded if (1) the trial was conducted in vitro or in vivo in an animal model or (2) the trial had a non-RCT design (e.g., case report, case series, or prospectively designed trial without a comparison group).

### 2.4. Outcome Measures

In this study, the primary outcomes of interest were measures of pain, muscle mass, and muscle strength. The muscle mass measures included but were not limited to lean body mass, fat-free mass, appendicular lean mass, muscle cross-sectional area (CSA), muscle volume, and muscle thickness. Pain was measured using a perceived-report scale, such as the visual analog scale.

The secondary outcomes of interest comprised physical mobility (specifically, walking capability, chair-rise performance, and timed up-and-go performance); physical activity; patient-reported global functioning; and levels of inflammatory biomarkers of OA, such as C-reactive protein (CRP), interleukin-6 (IL-6), and tumor necrosis factor-α (TNF-α) [43]. Specifically, walking capability was assessed using walking task outcomes, such as 10-m walk time and timed 6-min walk distance test outcomes. Patient-reported global functioning was assessed using a perceived function scale, such as the Harris Hip Score (HHS) [44], the Knee injury and Osteoarthritis Outcome Score (KOOS) [45], and the 36-Item Short-Form Health Survey (SF-36) physical score [46]. If improved patient condition is indicated by a negative score change for a measure (e.g., pain score and timed up-and-go performance), score changes for that measure had their signs inverted in the meta-analyses.

### 2.5. Data Extraction

Data on the following were extracted from each included trial and are presented in an evidence table: (1) characteristics of the study sample and research design, including group design and participant type; (2) characteristics of ET and PS interventions; (3) measured time points; and (4) main outcome measures. One author (CDL) extracted the relevant data from the included trials, and another author (HCC) reviewed the extracted data. Any disagreement between the two authors was resolved through discussion to reach a consensus. A third author (THL) was consulted if a consensus could not be reached.

### 2.6. Assessment of Risk of Bias and Methodological Quality of Included Studies

To assess the quality of the included trials, the PEDro quality score was estimated for assessing the risk of bias. Two researchers independently assessed the methodological quality of each of the included studies in accordance with the PEDro classification scale, which is a valid measurement tool of the methodological quality of clinical trials [47]. The PEDro scale comprises 10 items: random allocation, concealed allocation, similarity at baseline, subject blinding, therapist blinding, assessor blinding, >85% follow-up for at least one key outcome, intention-to-treat analysis, between-group statistical comparison for at least one key outcome, and point and variability measures for at least one key outcome. Each item is scored as either 1 (present) or 0 (absent), with the total score ranging from 0 to 10. The validity and reliability of the PEDro scale have been verified in previous studies: an inter-rater reliability generalized kappa statistic between 0.53 and 0.94 was reported [48], and an intraclass correlation coefficient of 0.91 [95% confidence interval (CI): 0.84–0.95] was found for the cumulative PEDro score in a nonpharmacological study [49]. In this study, the methodological quality of the included RCTs was rated as high (≥7), medium (4–6), or low (≤3) based on the total PEDro score (which is 10) [50].

### 2.7. Data Synthesis and Analysis

We computed effect sizes for each study separately for primary and secondary outcome measures. The primary and secondary outcome measures were defined as a pooled estimate of the mean difference in the change between the treatment (i.e., PS + ET) and control (i.e., ET with or without placebo supplementation) groups. If the exact variance of the paired difference was not derivable, the variance was imputed by assuming the within-participant correlation coefficients—between the baseline and post-test measured data—of 0.98 for muscle mass [51], 0.92 for muscle strength [52,53], 0.8 for mobility [53,54], and 0.5 for patient-reported pain and global functioning [55]. All extracted outcome data were transformed into the mean difference (MD) relative to the control. When different scales, we used the standardized MD (SMD) for meta-analysis were used to measure the same outcome (e.g., pain and muscle strength) [56]. Follow-up duration was assessed and classified into short-term (≤1 month), medium-term (>1 month and <6 months), and long-term (≥6 months) durations.

Fixed-effect or random-effect models were used depending on the presence of heterogeneity. Statistical heterogeneity was assessed using the *I*^2^ statistic, with significance at *p* < 0.05 [57]. A fixed-effect model was used unless statistical heterogeneity was significant, in which case a random-effect model was used.

The follow-up duration was assessed and defined as short term (≤1 month), medium term (>1 month and <6 months), and long term (≥6 months).

If heterogeneity was significant, subgroup analyses were further performed to explore the potential modifiers of PS + ET treatment effects [57]. The subgroup analyses considered the following: (1) participant-related factors, including sex, body mass index (≥30 kg/m^2^ or <30 kg/m^2^), and involved joint (hip or knee); (2) study design–related factors, including methodological quality (PEDro score ≥7 or <7); and (3) intervention-related factors, including PS dose (≥40 g/day or <40 g/day [58]) and duration of intervention. All subgroup differences were tested for significance, and the *I*^2^ statistic was also computed to estimate the degree of subgroup variability. Potential publication bias was investigated through the initial visual inspection of a funnel plot for reporting bias [59] and a subsequent Egger’s regression asymmetry test [60] using SPSS (Version 22.0, IBM, Armonk, NY, USA). Statistical significance was set at *p* < 0.05. All analyses were conducted using RevMan 5 software (Version 5.3, The Nordic Cochrane Centre, Copenhagen, Denmark).

## 3. Results

### 3.1. Selection Process of Studies

Figure 1 illustrates the flowchart of the selection process of studies. Through electronic and manual literature searches, we identified a total of 289 articles and removed any duplicates. We then reviewed the titles and abstracts of 83 studies to assess their eligibility, of which 26 were considered relevant to this study and thus underwent full-text assessment (Figure 1). The final sample comprised six RCTs [31,32,61,62,63,64] published between 2013 and 2019. All of the six RCTs were included in the qualitative synthesis, and five of them were included in the meta-analysis [31,32,61,62,63].

### 3.2. Study Characteristics

Table 1 summarizes the study characteristics and patient demographics of the included RCTs. The RCTs recruited a total of 242 participants with a mean age of 66.9 years (range: 63.6−75.6 years) and mean BMI of 28.1 kg/m^2^ (range: 21.9−34.0 kg/m^2^); 53.0% of the participants were female (range: 52.6%−77.3%), which was estimated after excluding two sex-specific (women-only) RCTs [32,64].

Among all participants, 123 (50.8%) received PS + ET, and 119 (49.2%) received ET with placebo PS. In addition, 112 (46.3%) patients enrolled in three RCTs received a diagnosis of knee OA and underwent total knee replacement (TKR) [31,61,62]. Furthermore, three RCTs enrolled 130 (53.7%) older adults with a diagnosis of hip OA [32,63,64], of which two RCTs enrolled patients who underwent total hip replacement (THR) [32,63]. All the included RCTs enrolled patients who had OA with a K-L grade of III or IV, which was classified as moderate or severe OA, respectively [42].

Regarding the duration of follow-up for measuring outcomes, all included RCTs had short-term follow-up (≤1 month), and four RCTs had medium-term follow-up (follow-up duration in these four RCTs: 7–16 weeks) [31,61,62,63]. None of the included RCTs reported long-term follow-up (≥6 months).

### 3.3. Protein Supplementation Characteristics

Protocols for PS are summarized in Table 1. The protocols for PS, including additional PS, are detailed in Appendix A. The protocol for the protein nutrient intervention varied widely across the included RCTs. Among the six included RCTs, four prescribed essential amino acids, such as leucine [31,61,62,63], and two prescribed branched chain amino acids [32,64] at a supplement dose of 3.0−6.0 g/session or 8.0−40.0 g/day. In particular, in five RCTs, PS interventions were administered to patients who received total joint replacement [31,32,61,62,63], with an intervention period of 2−7 weeks after surgery (Appendix A).

### 3.4. Exercise Training Protocol

The ET protocols are summarized in Table 1 and detailed in Appendix A. Regarding the training mode of exercise, one of the six RCTs used preoperative home-based RET for older women with OA who were scheduled to undergo primary unilateral total hip arthroplasty [64], and in the other five RCTs, postoperative physical therapy in combination with MET was provided to patients who underwent TKR or THR. All the included RCTs employed short-term treatment (2–7 weeks over 21–56 sessions), and none of the included RCTs employed long-term treatment (≥6 months).

### 3.5. Risk of Bias of Included Studies

The PEDro scores for individual studies are listed in Appendix A. Overall, three of the six included RCTs had high methodological quality [32,63,64], and the other three had medium methodological quality [31,61,62]; the median PEDro score was 6 (range: 6–9). The inter-rater reliability of the cumulative PEDro scores was acceptable, with an intraclass correlation coefficient of 0.97 (95% CI: 0.82–0.99). All the included RCTs had the PEDro methodological features of random allocation, similarity at the baseline, between-group comparisons, and point estimates and variability. However, all the three RCTs with high quality [32,63,64] employed allocation concealment in their methodology, but none of the RCTs with medium quality did. Furthermore, because of the nature of the interventions, therapist blinding was not plausible in some RCTs—although all RCTs with high and medium quality incorporated participant blinding into their PS intervention. In addition, assessor blinding was conducted in one [63] and three [31,61,62] of RCTs with high and medium quality, respectively.

### 3.6. Treatment Outcomes for Muscle Mass

Muscle mass outcomes were investigated by measuring mid-thigh muscle CSA [62], upper-arm muscle CSA [32], and mid-thigh muscle volume [31,61] (Table 1). Two RCTs measured quadriceps and hamstring muscle volumes in both the involved (i.e., operative) and contralateral uninvolved (i.e., nonoperative) legs, respectively, in patients who underwent primary unilateral TKR [31,61]. The combined analysis demonstrated that over medium-term follow-up, PS + ET significantly increased quadriceps muscle volume in both the involved leg (weighted MD = 7.38%, *p* < 0.0001; *I*^2^ = 71%; Appendix A) and uninvolved leg (weighted MD = 6.11%, *p* = 0.0002; *I*^2^ = 0%; Appendix A). Similarly, over medium-term follow-up duration, PS + ET improved hamstring muscle volume in both the operative leg (weighted MD = 7.71%, *p* < 0.00001; *I*^2^ = 65%; Appendix A) and uninvolved leg (weighted MD = 6.08%, *p* < 0.0001; *I*^2^ = 0%; Appendix A).

One included RCT assessed changes in upper-arm CSA resulting from PS + ET administered after THR surgery [32]; according to its results, over short-term follow-up, PS + ET prevented postoperative upper-arm muscle atrophy, with an MD of 1.4 cm^2^ (*p* < 0.05; Appendix A). Similarly, one other included RCT investigated changes in the number of quadriceps myofibers resulting from PS + ET administered after TKR surgery [62]; according to its results, over short-term follow-up, PS + ET significantly improved the number of such myofibers in the uninvolved leg (MD = 19.8, *p* < 0.01) but not the involved leg (Appendix A).

When all the muscle mass measures were pooled in this meta-analysis, the results indicated that PS + ET yielded significant improvements to muscle mass (SMD = 1.13, 95% CI: 0.72–1.53, *p* < 0.00001; *I*^2^ = 0%), favoring PS + ET for all follow-up durations (Figure 2).

### 3.7. Pain-Related Treatment Outcomes

Pain outcomes were assessed using perceived-reported scales, which included the HHS pain subscale in one RCT [63] and the KOOS pain subscale in another RCT [61] (Table 1). The combined analysis revealed that PS + ET significantly reduced pain: the SMD of 1.36 (95% CI: 0.68–2.03, *p* < 0.00001; *I*^2^ = 54%) was significant for the overall follow-up durations (Figure 3).

### 3.8. Treatment Outcomes for Muscle Strength

Three RCTs reported that PS + ET administered after total joint replacement improved lower-limb strength [31,32,61] (Table 1). The meta-analysis results revealed that over the overall follow-up period, PS + ET improved muscle strength in both the involved leg (SMD = 0.44, *p* = 0.04; *I*^2^ = 52%; Figure 4A) and the uninvolved leg (SMD = 0.54, *p* = 0.01; *I*^2^ = 0%; Figure 4B). One RCT administered PS in combination with home-based RET for older women with hip OA during their waiting period for primary unilateral THR [64]; according to its results, PS + ET significantly improved hip abductor muscle strength in the unaffected leg (MD = 16.8%, *p* < 0.01) but not the affected leg (Appendix A).

### 3.9. Treatment Outcomes Related to Physical Mobility and Physical Activity

Postoperative physical mobility was assessed for patients who underwent total joint replacement. Such assessment was based on several performance-based functional activity tasks, including walking capability tests in three RCTs [31,61,64], timed up-and-go tests in two RCTs [31,61], and stair climb tests in two RCTs [31,61] (Table 1). The meta-analysis revealed that over the overall follow-up durations, PS + ET improved walking capability (SMD = 0.75, *p* = 0.003; *I*^2^ = 0%; Figure 5a), timed up-and-go performance (SMD = 0.66, *p* = 0.01; *I*^2^ = 0%; Figure 5b), and stair climbing performance (SMD = 0.84, *p* = 0.002; *I*^2^ = 4%; Figure 5c). One RCT analyzed the 10-m walk time before and after PS + ET for older women with hip OA who did not receive THR [64]; the results revealed that the improvement rate of walking capability nonsignificantly differed between the PS + ET and control groups (Appendix A).

Over medium-term follow-up, postoperative physical activity was measured in terms of daily energy expenditure in one RCT [61] and daily step count in another RCT [31]. The meta-analysis revealed that PS + ET nonsignificantly improved physical activity (Figure 5d).

### 3.10. Treatment Outcomes in Patient-Reported Global Functioning

In total, two of the included RCTs reported global functional outcomes that were assessed using self-perception questionnaires: the HHS in one RCT [63] and the SF-36 physical function subscale in the other RCT [61] (Table 1). An analysis of the combined results from the RCTs revealed that PS + ET provided significantly greater improvements to global functioning relative to the control intervention: the SMD was 0.65 (95% CI: 0.20−1.10, *p* = 0.004; *I*^2^ = 17%), regardless of the involved joint and follow-up duration (Figure 5e).

### 3.11. Treatment Outcomes for Inflammatory Factors

Systemic concentrations of proinflammatory cytokines were measured; concentrations of CRP, IL-6, and TNF-α were measured in two RCTs [61,63], one RCT [62], and one RCT [62], respectively (Table 1). The meta-analysis revealed that PS + ET improved CRP outcomes (weighted MD = 0.16 mg/L, *p* = 0.04; *I*^2^ = 0%; Appendix A). Furthermore, changes in IL-6 and TNF-α levels did not significantly differ between the PS + ET and control groups (Appendix A).

When all the reported inflammatory measures were pooled in the meta-analysis, the results revealed an SMD of 0.37 (95% CI: 0.03–0.70, *p* = 0.03; *I*^2^ = 0%), favoring PS + ET, regardless of the follow-up period (Figure 6).

### 3.12. Side Effects and Compliance

All the included RCTs reported no clinically relevant adverse events, side effects, or major complications after the ET or PS intervention. Compliance to RET was reported to be 85.0%–88.2% in one RCT [64], and no RCT reported the compliance or attendance rate for MET sessions (Table 1). Compliance to PS was reported to be 83.4%–100% in three RCTs [32,62,64] (Table 1).

### 3.13. Publication Bias

The funnel plots of the effect sizes of each of the primary and secondary outcome measures are presented in Appendix A. For all measures, visual evaluation of their funnel plots revealed no substantial asymmetry. A subsequent Egger’s linear regression test also indicated no obvious reporting bias among the comparisons for all measures (*p* > 0.05).

## 4. Discussion

This study demonstrated that (1) PS + ET significantly improves muscle mass, muscle strength, and pain outcomes in older adults with lower-extremity OA, regardless of the follow-up duration, and that (2) PS + ET significantly improves secondary clinical outcomes, including physical mobility, systemic inflammation, and patient-reported global functioning. Subgroup analyses were not performed because all outcome measures exhibited no significant heterogeneity.

Previous systematic reviews and meta-analyses have investigated the effects of PS + ET on muscle mass outcomes in older adults who either were frail or had sarcopenia; these studies have reported—relative to ET alone—overall effects of PS + ET on the weighted MDs of lean mass gain ranging from 0.41 to 1.60 kg [29,65] or SMDs ranging from 0.23 to 2.05 [38,66,67,68,69] (Table 2). The present meta-analysis noted that PS + ET exerted greater effects on changes in muscle mass (SMD = 1.13) relative to the control; this finding indicates that PS + ET reverses or prevents the loss of muscle mass in older adults with OA. Our findings support the urgent necessity of incorporating protein-based nutrition or dietary intervention into ET to prevent functional decline in older adults with OA at risk of sarcopenia; this intervention is especially urgent for such older adults who have recently undergone total knee or hip replacement. This recommendation agrees with previous systematic reviews and with recommendations proposed by the European Society for Clinical Nutrition and Metabolism Expert Group [70].

The present study noted that in older adults with OA, PS + ET significantly improved muscle mass; this finding suggests that PS further augments muscle-mass increases from exercise interventions. This finding is consistent with those of previous systematic reviews on the effectiveness of PS + ET in sarcopenic or frail older adults [38,68], regardless of exercise type. Some reasons potentially explain the findings of the present study. First, both PS (especially with leucine and whey protein [71,72]) and ET stimulate myoprotein synthesis in older adults [71,73,74]; such stimulation occurs despite delayed and diminished muscle protein anabolic responses to PS + ET in older adults relative to younger controls [75,76]. In addition, in response to PS, older men with sarcopenia show an elevated rate of muscle protein synthesis—at a level comparable to that of their healthy peers [74]. The rate of muscle protein synthesis is more significantly increased by PS + ET than by either ET alone [77] or PS alone [78,79], which also explains the results of previous studies [38,68]. Second, muscle wasting or atrophy in OA is primarily mediated by pain as well as by OA-associated inflammation through the disease process [7]. Loss of muscle mass in chronic diseases is frequently associated with increased production of proinflammatory cytokines (such as TNF-α and IL-6) and acute-phase inflammatory proteins (such as CRP), all of which are inflammatory biomarkers of OA [43]. Proinflammatory cytokines act on muscle protein metabolism by not only activating catabolic pathways but also downregulating the anabolic pathways; accordingly, systemic inflammation is associated with decreased rates of myoprotein synthesis accompanied with enhanced protein degradation (i.e., muscle breakdown) [80]. Therefore, treatments that are aimed at reducing levels of systemic inflammation may increase the rate of muscle mass gain. The present study revealed that PS + ET had a favorable effect on the reduction of systemic inflammatory biomarker concentrations, which potentially explains the muscle mass outcomes in this study.

The use of an additional PS product may further increase lean mass in patients with OA; we surmise this because a recent meta-analysis indicated that muscle strengthening exercises benefit muscle hypertrophy and lean mass gain in patients with OA [81]. However, in contrast to the results of the present study, previous systematic reviews have noted nonsignificant effects of PS + ET (usually RET), relative to ET alone, on changes in muscle mass in older adults who mostly were healthy or without OA [67,82,83]. This inconsistency in results is attributable to the differences in study populations. Furthermore, the differences in findings between previous reviews and the present meta-analysis further confirm the conclusion in the literature that older people who have either sarcopenia alone or sarcopenia with additional medical conditions may more greatly benefit from PS + ET than their healthy peers do in terms of lean mass gain and physical performance [29,84]. Therefore, targeting muscle mass through PS + ET administration serves as a promising intervention for preserving independence in daily living activities and preventing disease progression in older adults with OA.

Our study has several limitations. First, because of the variation in PS regimes (with respect to protein source, supplied amounts, and timing of ingestion) and ET regimes (with respect to training duration and training volume), our study did not yield a definite conclusion regarding the effect of a specific type of PS or ET on muscle mass or strength gain. Second, some of our included trials had small sample sizes [31,32]; these studies’ results have indicated no significant intervention effect on primary or secondary outcomes, which potentially diminished the overall effect size. Finally, none of the included RCTs reported long-term follow-up (≥6 months) outcomes for muscle mass and functioning. Thus, future studies should determine the long-term effects of PS + ET over follow-up periods ≥6 months.

## 5. Conclusions

This meta-analysis demonstrated that PS is effective as a nutritional intervention; it yields improvements in muscle mass and strength during postoperative rehabilitation (mostly in MET regimes) for older adults with lower-extremity OA who have undergone total joint replacement. Postoperative PS further reduces pain, increases physical mobility, and improves global functioning after 2−7 weeks of rehabilitative ET. Considering the small number of RCTs included in this meta-analysis, more future studies are required to more robustly determine the efficacy of PS + ET in this specific population. In addition, only one RCT enrolled patients without prosthesis in our meta-analysis; therefore, we could not definitively conclude that PS + ET is effective in older adults with OA who did not undergo total joint replacement. Thus, future studies on the effectiveness of PS + ET should focus on people with OA who have not undergone total joint replacement. The results elucidate nutritional and exercise interventions that benefit older adults with OA, particularly those who have undergone total joint replacement. The results also facilitate the formulation of practical and interdisciplinary approaches to counteracting muscle loss and functional decline. Practitioners in geriatric care and in rehabilitation settings, such as clinics, hospitals, institutions, and communities, can use our findings as a reference. Nevertheless, to better identify more optimal supplementation protocols, our results must be further validated by additional studies with relatively large samples.

## Figures and Tables

**Figure 1 nutrients-12-02422-f001:**
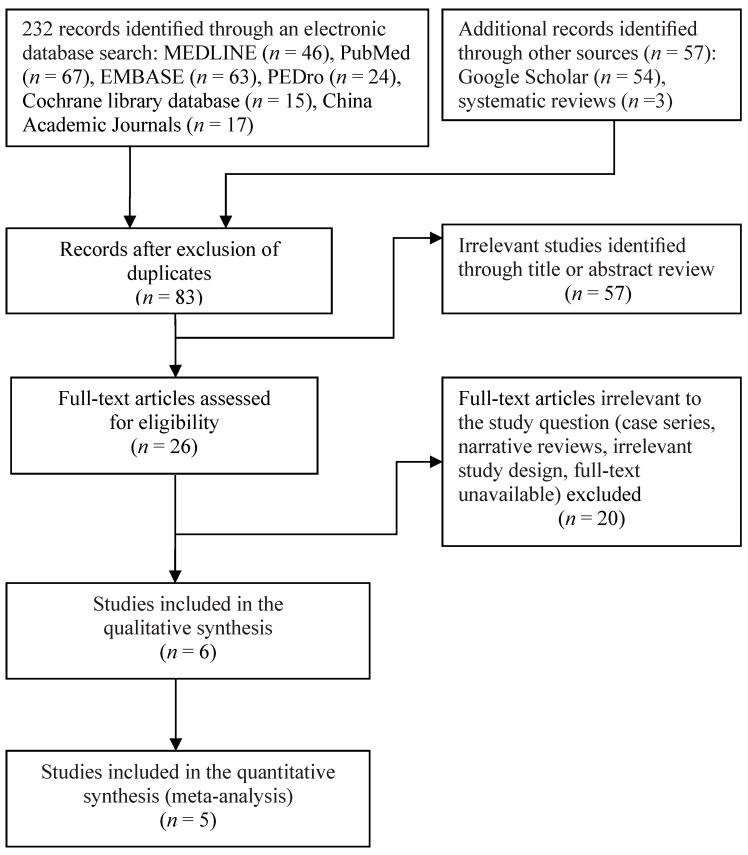
Flowchart of enrollment of the studies.

**Figure 2 nutrients-12-02422-f002:**
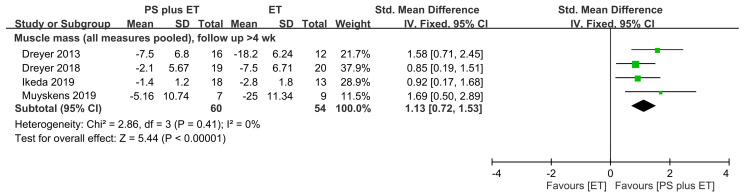
Forest plot summarizing the effects of postoperative protein supplementation (PS) plus exercise training (ET) on changes in muscle mass over medium-term follow-up. The square represents the point estimate of the intervention effect for each trial. The horizontal line links the lower and upper limits of the 95% CI for the given effect. The area of the squares indicates the relative weight of the trials in the meta-analysis. Trial results plotted on the right-hand side of the vertical axis indicate effects in favor of protein supplementation. The combined effects are plotted using black diamonds. 95% CI, 95% confidence interval; Std, standard; IV, inverse variance.

**Figure 3 nutrients-12-02422-f003:**
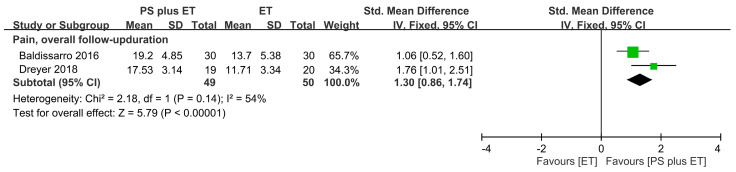
Forest plot summarizing the effects of postoperative protein supplementation (PS) plus exercise training (ET) on changes in the pain score over the overall follow-up period. The square represents the point estimate of the intervention effect for each trial. The horizontal line links the lower and upper limits of the 95% CI for the given effect. The area of the squares indicates the relative weight of the trials in the meta-analysis. Trial results plotted on the right-hand side of the vertical axis indicate effects in favor of protein supplementation. The combined effects are plotted using black diamonds. 95% CI, 95% confidence interval; Std, standard; IV, inverse variance.

**Figure 4 nutrients-12-02422-f004:**
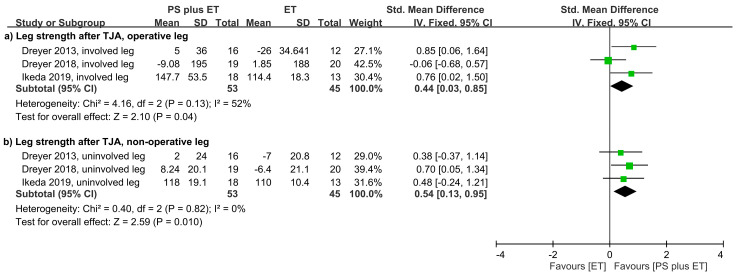
Forest plot summarizing the effects of postoperative protein supplementation (PS) plus exercise training (ET) on changes in muscle strength of (**a**) the involved leg and (**b**) the contralateral uninvolved leg over the overall follow-up period. The square represents the point estimate of the intervention effect for each trial. The horizontal line links the lower and upper limits of the 95% CI for the given effect. The area of the squares indicates the relative weight of the trials in the meta-analysis. Trial results plotted on the right-hand side of the vertical axis indicate effects in favor of protein supplementation. The combined effects are plotted using black diamonds. 95% CI, 95% confidence interval; Std, standard; IV, inverse variance; TJA, total joint arthroplasty.

**Figure 5 nutrients-12-02422-f005:**
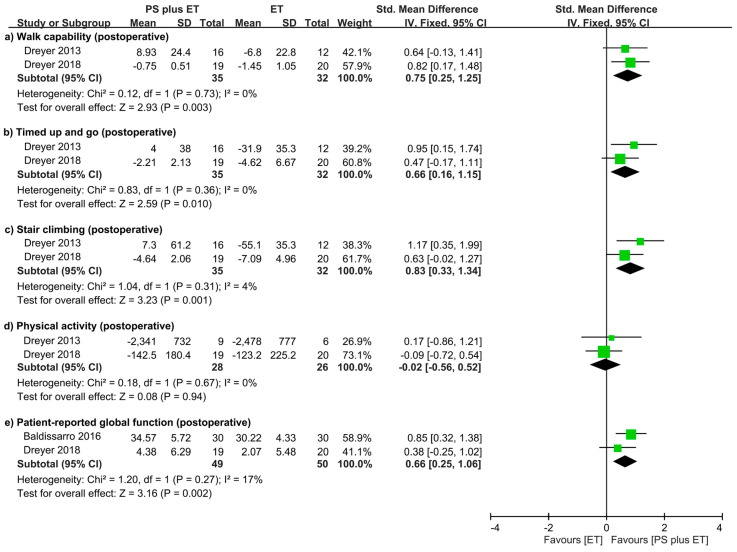
Forest plot summarizing the effects of postoperative protein supplementation (PS) plus exercise training (ET) on changes in (**a**) walking capability; (**b**) timed up-and-go performance; (**c**) stair climbing performance; (**d**) physical activity; and (**e**) global functioning over the overall follow-up period. The square represents the point estimate of the intervention effect for each trial. The horizontal line links the lower and upper limits of the 95% CI for the given effect. The area of the squares indicates the relative weight of the trials in the meta-analysis. Trial results plotted on the right-hand side of the vertical axis indicate effects in favor of protein supplementation. The combined effects are plotted using black diamonds. 95% CI, 95% confidence interval; Std, standard; IV, inverse variance.

**Figure 6 nutrients-12-02422-f006:**
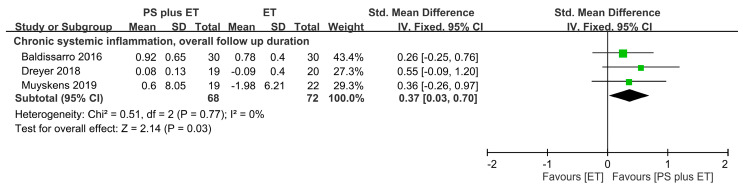
Forest plot summarizing the effects of postoperative protein supplementation (PS) plus exercise training (ET) on changes in systemic inflammation level during an overall follow-up duration. For each trial, the square represents the point estimate of the intervention effect. The horizontal line links the lower and upper limits of the 95% CI of this effect. The area of the squares reflects the relative weight of the trials in the meta-analysis. Trial results plotted on the right-hand side of the vertical axis indicate effects in favor of protein supplementation. The combined effects are plotted using black diamonds. 95% CI, 95% confidence interval; Std, standard; IV, inverse variance.

**Table 1 nutrients-12-02422-t001:** Characteristics of the included studies.

Study (Author, Year, ref)	Group ^a^	N	Design	Patient Type	Exercise Intervention		Dietary Intervention		Measured Time Point	Outcome Results
					Type, (Compliance: %, EG/CG) ^b^	Frequency × Duration	Mode, Type (Compliance: %, EG/CG) ^b^	PS Dose (g/day or g/session)		
Baldissarro, 2016 [63]	EG: PS + ET	30	RCT	THA	PostOP MET	2 session/d × 2 wk	PS, EEA	8 g/day	Baseline	Harris Hip Score
	CG: PLA + ET	30			(NR)	(24 sessions)	(NR)		Posttest: 1, 2, 8, 16 wk	CRP
Dreyer, 2013 [31]	EG: PS + ET	16	RCT	TKA	PostOP MET	2 session/d × 2 wk	PS, EAA	40 g/day	Baseline	Qd volume
	CG: PLA + ET	12	DB		(NR)	(24 sessions)	(NR)		Posttest: 3 wk	Qd strength; PA
									Follow-up: 7 wk	TUG; SC; 6MWD
Dreyer, 2018 [61]	EG: PS + ET	19	RCT	TKA	PostOP MET	2 session/d × 2 wk	PS, EAA	40 g/day	Baseline	Qd strength; CRP
	CG: PLA + ET	20	DB		(NR)	(24 sessions)	(NR)		Posttest: 3 wk	Qd volume; SC; GS
									Follow-up: 7 wk	TUG; SF-36 PF; PA
Ikeda, 2018 [64]	EG: PS + ET	21	RCT	Hip	RET	7 d/wk × 4 wk	PS, BCAA	6.0 g/session	Baseline	GS
	CG: PLA + ET	22	SB	OA	(85.0/88.2)	(28 sessions)	(83.4/92.0)		Posttest: 4 wk	Hip strength
Ikeda, 2019 [32]	EG: PS + ET	18	RCT	THA	PostOP MET	2 session/d × 4 wk	PS, BCAA	3.0 g/session	Baseline	Upper arm CSA
	CG: PLA + ET	13	SB		(NR)	(56 sessions)	(100/100)		Posttest: HIDC	Qd strength
Muyskens, 2019 [62]	EG: PS + ET	19	RCT	TKA	PostOP MET	2–3 d/wk × 7 wk	PS, EAA	40.0 g/day	Baseline	Number of myo-
	CG: PLA + ET	22	DB		(NR)	(21 sessions)	(99/96)		Mid-test: 2, 3 wk	fibers (Qd)
									Posttest: 7 wk	IL-6; TNF-α

^a^ All parallels of experimental and control groups are presented for each trial. ^b^ Values denote the compliance of interventions (%). 6MWD, 6-min walk-for-distance; BCAA, branched chain amino acids; CG, control group; CSA, cross-sectional area; DB, double blind; EAA, essential amino acids; EG, experimental group; ET, exercise training; GS, gait speed; HIDC, hospital inpatient discharge; IL-6, interleukin-6; MET, multicomponent exercise training; NR, not reported; PA, physical activity; PLA, placebo supplement; PS, protein supplementation; Qd, quadriceps muscle; RCT, randomized controlled trial; ref, reference number; RET, resistance exercise training; SB, single blind; SC, stair climbing; SF-36 PF, Short-Form 36-Item Health Survey physical function subscore; TKA, total knee arthroplasty; TNF-α, tumor necrosis factor-α; THA, total hip arthroplasty; TUG, timed up and go test; wk week.

**Table 2 nutrients-12-02422-t002:** Meta-analysis results for effects of protein supplementation plus exercise on muscle mass in older adults.

Study (Author, Year, Reference)	Participant Characteristics		Intervention Design			Outcome (favoring PS + ET)
			EG (PS + ET)		CG	Muscle Mass ^a^
	Population	Age (years)	ET	PS	Comparison	
Finger, 2015 [67]	Sarcopenia	60−79	RET, 2−3 days/week, 12−72 weeks	Whey, milk protein, EAA (6−40 g/day)	RET alone or with placebo supplement	FFM: SMD = 0.23 (0.05, 0.42)
Liao, 2017 [38]; 2018 [69]	Overweight and obesity; sarcopenia	60−85	RET, 2−7 days/week, 12−24 weeks	Whey, leucine (10–35 g/day)	RET alone or with placebo supplement	LBM: SMD = 0.52 (0.18, 0.85)
Luo, 2017 [66]	Sarcopenia	65−80	RET or MET, 2−3 days/week, 12−16 weeks	Whey (20−40 g/day), EAA (6−17 g/day)	ET alone	FFM: SMD = 5.78 (5.17, 6.40) LBM: SMD = 2.05 (0.91, 3.19)
Hidayat, 2018 [29]	Chronic conditions	60−80	RET, 3−5 days/week, 12−72 weeks	Whey, milk protein (13−40 g/day)	RET with placebo PS or low-protein diet	FFM: WMD = 1.60 (0.92, 2.28) kg
Hita-Contreras, 2018 [65]	Sarcopenia and obesity	76−81	RET, 1−2 days/week, 12−26 weeks	Whey (40 g/day), EAA (leucine, 3 g/day)	Regular care (non-ET, non-PS)	ALM: WMD = 0.41 (0.68, 0.65) kg SMI: SMD = 0.47 (−0.1, 1.04)
Liao, 2019 [68]	Sarcopenia or frailty	64−89	RET or MET: 2−7 days/week, 3−36 weeks	Whey, milk protein, EAA (3−40 g/day)	ET alone (with or without placebo supplement)	LBM: SMD = 0.53 (0.21, 0.86) ALM: SMD = 0.40 (−0.02, 0.52)
Current study	Osteoarthritis	63−76	RET or MET: 2−7 d/week, 2−24 weeks	EAA, BCAA (3−40 g/day)	ET alone (with or without placebo supplement)	Muscle mass: SMD = 1.13 (0.72, 1.53)

^a^ Data are presented as WMD or SMD with 95% confidence interval in parentheses. ALM, appendicular lean mass; BCAA, branched chain amino acids; CG, control group; EAA, essential amino acid; EG, experimental group; ET, exercise training; FFM, fat-free mass; LBM, lean body mass; MET, multicomponent exercise training; PS, protein supplementation; RET; resistance exercise training; SMD, standardized mean difference: SMI, skeletal mass index; WMD, weighted mean difference.

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
