# Peer review of "Effects of Protein Supplementation Combined with Exercise Training on Muscle Mass and Function in Older Adults with Lower-Extremity Osteoarthritis: A Systematic Review and Meta-Analysis of Randomized Trials"

_nutrients, 2020, doi:10.3390/nu12082422_

Round 1

Reviewer 1 Report

Overall considerations

Manuscript titled “Effects of Protein Supplementation Combined with Exercise Training on Muscle Mass and Function in Older Adults with Lower Extremity Osteoarthritis: A Systematic Review and Meta-analysis of Randomized Trials” deal an important issue of OA. This study determined the effectiveness of protein supplementation plus exercise 22 training (PS+ET) in alleviating muscle mass and function outcomes in elderly patients with lower-limb OA. These findings suggest that PS+ET exerts overall benefits on muscle mass and clinical outcome in elderly adults with lower-limb OA who have a high risk of sarcopenia.

This paper is interesting and suitable with the purpose of the journal even if major and minor concerns need to be solved before suggesting for publication.

Authors should consider making the paper clearer. It is full of SMD and P value which distract the reader too much. The aim of the study sometimes seems to be forgotten due to the presence of too many data. The project should be streamlined.

The introduction section is not updated based on the current literature.

In the introduction you could add, for example, a sentence regarding the importance of physical activity and nutrition as non-pharmacological and preventive treatment for OA and sarcopenia. Authors can comment and quote the following interesting and recent papers as follow:

Physical activity and Mediterranean diet based on olive tree phenolic compounds from two different geographical areas have protective effects on early osteoarthritis, muscle atrophy and hepatic steatosis. Eur J Nutr. 2019 Mar;58(2):565-581. doi: 10.1007/s00394-018-1632-2. Epub 2018 Feb 15. PMID: 29450729.

Impact of Western and Mediterranean Diets and Vitamin D on Muscle Fibers of Sedentary Rats. Nutrients. 2018 Feb 17;10(2):231. doi: 10.3390/nu10020231. PMID: 29462978; PMCID: PMC5852807.

Assessment of Vitamin D Supplementation on Articular Cartilage Morphology in a Young Healthy Sedentary Rat Model. Nutrients. 2019 Jun 3;11(6):1260. doi: 10.3390/nu11061260. PMID: 31163658; PMCID: PMC6628271.

Moderate Physical Activity as a Prevention Method for Knee Osteoarthritis and the Role of Synoviocytes as Biological Key. Int J Mol Sci. 2019 Jan 25;20(3):511. doi: 10.3390/ijms20030511. PMID: 30691048; PMCID: PMC6387266.

A mention of conservative management could make the article interesting for those working in the rehabilitation field like the use of kinesio tape or the different type of exercise intervention on sarcopenia. Please consider to comment and quote the following papers:

The Effects of Exercise and Kinesio Tape on Physical Limitations in Patients with Knee Osteoarthritis. J. Funct. Morphol. Kinesiol. 2016, 1, 355-368.

Sarcopenia and Exercise “The State of the Art”. J. Funct. Morphol. Kinesiol. 2017, 2, 40.

The manuscript

41 It is too general, the authors have to be more detailed about which grade of OA involves deficits in muscle volume and function.

62-64 I’m not sure about “unclear whether PS in combination with ET (PS+ET) exerts any effect on muscle mass,” because a similar study reported a significant improvement in hip abductor muscle strength in patients with OA. [Ikeda T, Jinno T, Masuda T, et al. Effect of exercise therapy combined with branched-chain amino acid supplementation on muscle strengthening in persons with osteoarthritis. Hong Kong Physiother J. 2018;38(1):23-31. doi:10.1142/S1013702518500038]

64 “Because of that” that what? There are no previous terms that show the increased risk of sarcopenia in older individuals.

71-73 Is a repetition of 62-64 contents

107 There is a repetition of number (7) while it has to be number (8)

219 The Kellgren-Lawrence classification need to be explained correctly since it is a tool to estimate the OA grade. Readers have to understand immediately that authors are talking about K-L grade III or IV like at line 41 where the definition of OA is a bit shallow.

226 Authors should reconsider the RTCs collected for the paper. There are 6 studies about the supplementation of additional proteins (229) and 7 studies about a high-protein diet (232).  The effects of additional proteins, namely a well-defined dose of the drug, cannot be compared with a high-protein diet where the dose of eaten protein is not clearly defined.

236 Authors should reconsider the RTCs collected for the paper. Data cannot be compared due to the different outcomes between the 3 RCTs with a medium duration (24 weeks) of treatment versus the 12 other RCTs with a short duration (2-7 weeks) of treatment. Since 12 RCTs are enough, I suggest discussing only about these ones.

302-304 The authors should clarify these data to avoid confusion.

324 If authors are talking only about OA of lower limbs, to mention a handgrip is not relevant.

406-407 What authors mean with “the outcomes of muscle strength as well as systemic inflammation were not affected by any factor defined in this study”? The muscle strength is directly linked to the different exercise protocol used by the RCTs.

465-467 If authors cannot compare the difference in treatment efficacy between male and female, cannot talk about how the difference in sex could influence the efficacy of treatments.

474 “Thus, etc...” messy, consider to make it clearer.

481-483 Authors are talking about a high intensity of training (80-95% of 1RM) which is not possible for OA patients; so it would be better if authors might hint a valid intensity of training worth for OA patients.

492-493 Comparing patients with total joint replacement versus those who did not have arthroplasty is not fair. The authors should consider to divide the pool of RCTs between prosthesis and non-prosthesis patients because the outcomes of PS+ET in joint replacement patients are surely different compared to those without arthroplasty. 

516 The 105 citation does not exist, authors have to adjust it.

Authors should highlight the outcomes of the study which seems to be difficult to understand immediately.

In the conclusion section please add limitations of the study and please highlight better the scientific/clinical relevance of your work. Please provide a clear message of the importance of this paper in the scientific community.

Tables

Table 1 is too messy. Authors should consider to make it clearer, maybe too many items are reported. If it is just a summary of included studies, items like age, BMI, sex are not needed.

Figure 1 has to be fixed correctly.

Table 3 is too messy, authors should consider to make it clearer.

Table S2 could be divided in two sections. One concerning the protein supplementation, one concerning the protein diet. As reported for line 226, authors should reconsider the included studies because they seem too different from each other.

Table S3 There is a bit of confusion among the included studies. I mean, almost the whole paper talks about exercise training where the patient follows a training session of several weeks. However, among the included studies there are some of them about physiotherapy rehabilitation while some others about a training session like strength training. The first ones requires the therapist's guidance, the second ones can be performed independently by the patient. Authors should consider to review the selection of studies in order to make a collection of more homogeneous studies, as reported for line 236.

Author Response

Reviewer 1

Comments and Suggestions for Authors

Overall considerations

Manuscript titled “Effects of Protein Supplementation Combined with Exercise Training on Muscle Mass and Function in Older Adults with Lower Extremity Osteoarthritis: A Systematic Review and Meta-analysis of Randomized Trials” deal an important issue of OA. This study determined the effectiveness of protein supplementation plus exercise 22 training (PS+ET) in alleviating muscle mass and function outcomes in elderly patients with lower-limb OA. These findings suggest that PS+ET exerts overall benefits on muscle mass and clinical outcome in elderly adults with lower-limb OA who have a high risk of sarcopenia.

This paper is interesting and suitable with the purpose of the journal even if major and minor concerns need to be solved before suggesting for publication.

Authors should consider making the paper clearer. It is full of SMD and P value which distract the reader too much. The aim of the study sometimes seems to be forgotten due to the presence of too many data. The project should be streamlined.

Response

We thank all the reviewers for their comprehensive review and their comments regarding our manuscript. We have made all necessary modifications to our originally submitted manuscript (nutrients-882515), based on reviewers’ comments, point by point.

In addition, our revised manuscript has been edited by a professional editing company (Wallace Academic Editing).

The introduction section is not updated based on the current literature.

In the introduction you could add, for example, a sentence regarding the importance of physical activity and nutrition as non-pharmacological and preventive treatment for OA and sarcopenia. Authors can comment and quote the following interesting and recent papers as follow:

Physical activity and Mediterranean diet based on olive tree phenolic compounds from two different geographical areas have protective effects on early osteoarthritis, muscle atrophy and hepatic steatosis. Eur J Nutr. 2019 Mar;58(2):565-581. doi: 10.1007/s00394-018-1632-2. Epub 2018 Feb 15. PMID: 29450729.

Impact of Western and Mediterranean Diets and Vitamin D on Muscle Fibers of Sedentary Rats. Nutrients. 2018 Feb 17;10(2):231. doi: 10.3390/nu10020231. PMID: 29462978; PMCID: PMC5852807.

Assessment of Vitamin D Supplementation on Articular Cartilage Morphology in a Young Healthy Sedentary Rat Model. Nutrients. 2019 Jun 3;11(6):1260. doi: 10.3390/nu11061260. PMID: 31163658; PMCID: PMC6628271.

Moderate Physical Activity as a Prevention Method for Knee Osteoarthritis and the Role of Synoviocytes as Biological Key. Int J Mol Sci. 2019 Jan 25;20(3):511. doi: 10.3390/ijms20030511. PMID: 30691048; PMCID: PMC6387266.

Response

Thank you for the constructive comment. In the revised manuscript, we added statements in the introduction section and cited the recommended references as follows:

Lines 58–59.

“Several recent studies have indicated physical activity and nutrition as nonpharmacological and preventive treatments for OA and sarcopenia [15-19].”

A mention of conservative management could make the article interesting for those working in the rehabilitation field like the use of kinesio tape or the different type of exercise intervention on sarcopenia. Please consider to comment and quote the following papers:

The Effects of Exercise and Kinesio Tape on Physical Limitations in Patients with Knee Osteoarthritis. J. Funct. Morphol. Kinesiol. 2016, 1, 355-368.

Sarcopenia and Exercise “The State of the Art”. J. Funct. Morphol. Kinesiol. 2017, 2, 40.

Response

In accordance with the reviewer’s comment, we added statements in the introduction section and cited the recommended references as follows:

Lines 60–63.

“in addition, conservative ET incorporated with varieties of training tools such as kinesio tape [21] and whole-body vibration (or electromyostimulation) [19], has been effectively employed to improve muscle mass, strength, and physical function [19, 22].”

The manuscript

41 It is too general, the authors have to be more detailed about which grade of OA involves deficits in muscle volume and function.

Response

We revised the statements as follows:

Lines 43–45.

“Deficits in muscle volume and function have been observed in older individuals with mild to moderate severity of hip OA [4] and those with mild and severe knee OA [5]; such deteriorations of muscle mass occurring with disease progression in OA have been attributed to sarcopenia, ……”

62-64 I’m not sure about “unclear whether PS in combination with ET (PS+ET) exerts any effect on muscle mass,” because a similar study reported a significant improvement in hip abductor muscle strength in patients with OA. [Ikeda T, Jinno T, Masuda T, et al. Effect of exercise therapy combined with branched-chain amino acid supplementation on muscle strengthening in persons with osteoarthritis. Hong Kong Physiother J. 2018;38(1):23-31. doi:10.1142/S1013702518500038]

Response

We revised the statements as follows:

Lines 68–72.

“However, results of among the previous studies regarding effectiveness of PS in combination with ET (PS+ET) for older individuals with OA or those who recently underwent total joint replacement remain inconsistency. Some previous studies observed that PS+ET achieved significant effects on muscle mass [30] and strength gains [31, 32] for elder individuals with OA whereas other authors had conflict results [33, 34].”

64 “Because of that” that what? There are no previous terms that show the increased risk of sarcopenia in older individuals.

Response

We revised the statements as follows:

Lines 72–74.

“Based on the facts that older individuals with OA have a high risk of sarcopenia [7], which may further affect postoperative outcomes in those who have recently received total joint replacement [35-37],”

71-73 Is a repetition of 62-64 contents

Response

We removed the statement which is indicated above.

107 There is a repetition of number (7) while it has to be number (8)

Response

We revised the statements as follows:

Lines 113–116.

“(7) PS+ET was applied either preoperatively or postoperatively to participants who underwent arthroplasty or replacement surgery in the knee or hip joint; and (8) the study reported at least one of the primary outcome measures including muscle mass, pain, and strength which were defined in the following.”

219 The Kellgren-Lawrence classification need to be explained correctly since it is a tool to estimate the OA grade. Readers have to understand immediately that authors are talking about K-L grade III or IV like at line 41 where the definition of OA is a bit shallow.

Response

We revised the statements to clarify the Kellgren-Lawrence classification for disease severity of OA as follows:

Lines 214–215.

“All the included RCTs enrolled patients who had OA with a K-L grade of III or IV, which were referred to moderate or severe OA, respectively [44].”

226 Authors should reconsider the RTCs collected for the paper. There are 6 studies about the supplementation of additional proteins (229) and 7 studies about a high-protein diet (232). The effects of additional proteins, namely a well-defined dose of the drug, cannot be compared with a high-protein diet where the dose of eaten protein is not clearly defined.

Response

Based on the reviewer’s comments, we revised to include RCTs about the supplementation of additional proteins (PS). We excluded the RCTs conducting a high-protein diet or supplementation of beta-hydroxy-beta-methylbutyrate. Accordingly, the keywords for intervention in search strategy were revised as follows:

Lines 99–101.

“The following keywords were used for the intervention: …… and (“protein/amino-acid/nutrient supplement”).”

Finally, a total of 6 RCTs regarding PS plus exercise were included in the revised manuscript, and the section of study results was also revised in accordance with the newly included RCTs. All tables and figures as well as supplementary tables and figures were revised accordingly. Please refer to the Results section.

236 Authors should reconsider the RTCs collected for the paper. Data cannot be compared due to the different outcomes between the 3 RCTs with a medium duration (24 weeks) of treatment versus the 12 other RCTs with a short duration (2-7 weeks) of treatment. Since 12 RCTs are enough, I suggest discussing only about these ones.

Response

In the revised manuscript, we reconsidered the RCTs collected for meta-analysis. As described above, 6 RCTs were included for re-analysis, all of which had a short treatment duration of 2-7 weeks.

302-304 The authors should clarify these data to avoid confusion.

Response

The subgroup analysis was not performed due to that no significant heterogeneity was identified in all outcome measures. Therefore, we removed the statements about results and discussions of subgroup analyses throughout the revised manuscript.

324 If authors are talking only about OA of lower limbs, to mention a handgrip is not relevant.

Response

In the revised manuscript, we removed the outcome measure, handgrip strength.

406-407 What authors mean with “the outcomes of muscle strength as well as systemic inflammation were not affected by any factor defined in this study”? The muscle strength is directly linked to the different exercise protocol used by the RCTs.

465-467 If authors cannot compare the difference in treatment efficacy between male and female, cannot talk about how the difference in sex could influence the efficacy of treatments.

474 “Thus, etc...” messy, consider to make it clearer.

481-483 Authors are talking about a high intensity of training (80-95% of 1RM) which is not possible for OA patients; so it would be better if authors might hint a valid intensity of training worth for OA patients.

Response

The subgroup analysis was not performed due to that no significant heterogeneity was identified in all outcome measures. Therefore, we removed the statements about results and discussions of subgroup analyses throughout the revised manuscript. We made statements to clarify it in Discussion section as follows:

Lines 361–363.

“The subgroup analyses were not performed due to that no significant heterogeneity was identified in all outcome measures.”

492-493 Comparing patients with total joint replacement versus those who did not have arthroplasty is not fair. The authors should consider to divide the pool of RCTs between prosthesis and non-prosthesis patients because the outcomes of PS+ET in joint replacement patients are surely different compared to those without arthroplasty.

Response

Following the reviewer’s comment, we separately performed meta-analysis for prosthesis and non-prosthesis patients. In the revised manuscript, only one RCT recruited non-prosthesis patients, which was not included in quantitative analysis. Therefore, only prosthesis patients enrolled by 5 RCTs were included in meta-analysis. Accordingly, the results and its discussions were revised throughout the revised manuscript.

516 The 105 citation does not exist, authors have to adjust it.

Response

The original 105 citation was removed in the revised manuscript.

Authors should highlight the outcomes of the study which seems to be difficult to understand immediately.

Response

We made statements to clarify outcomes of the study as follows:

Method section, 2.4. Outcome Measures

Line 121.

“Primary outcomes of interest in this study included measures of muscle mass, pain, and strength.”

Discussion section

Lines 358–361.

“This study demonstrated that PS+ET exerted significant effects on muscle mass, muscle strength, and pain in elderly people with lower-extremity OA, irrespective of the follow-up duration. In addition, PS+ET achieved significant effects on the secondary clinical outcomes in this study including physical mobility, patient-reported global function, and systemic inflammation.”

In the conclusion section please add limitations of the study and please highlight better the scientific/clinical relevance of your work. Please provide a clear message of the importance of this paper in the scientific community.

Response

We made statements about limitations of the study as follows:

Lines 410–418.

“Several limitations to our findings should be elucidated. First, because of the variation in PS regimes (protein source, supplied amounts, and timing of ingestion) and ET regimes (training duration and training volume), drawing a definite conclusion regarding the effect of a specific type of PS or ET on muscle mass or strength gain was difficult. Second, some of our included trials had small sample sizes [31, 32]; the results of these studies indicated no significant intervention effect on primary or secondary outcomes, which may have contributed negatively to the overall effect size. Finally, none of the included RCTs reported long-term follow up (≥6 months) outcomes for muscle mass and function; future studies are warranted to determine long-term effects of PS+ET during a follow-up period of ≥6 months.”

We made statements to provide the clinical relevance and the implications for further studies as follows:

Lines 420–430.

“This meta-analysis evidenced that PS displays an effective nutrition intervention to augment muscle mass and strength gains during postoperative rehabilitation (mostly MET regime) in elderly adults with lower-extremity OA who underwent total joint replacement. Postoperative PS further reduces pain, increases physical mobility, and improves global function after 2-7 weeks of rehabilitative ET. Considering the small number of RCTs included in this meta-analysis, more future studies are needed to establish the actual and robust efficacy of combined intervention PS and ET in such specific population. In addition, only 1 RCT which enrolled non-prosthesis patients was available in this study; therefore, we were not allowed to draw a solid conclusion regarding the efficacy of PS+ET for older adults with OA who did not undergo total joint replacement and future studies are warranted to investigate the effectiveness of PS+ET for OA population without total joint replacement.”

Tables

Table 1 is too messy. Authors should consider to make it clearer, maybe too many items are reported. If it is just a summary of included studies, items like age, BMI, sex are not needed.

Response

According to the reviewer’s constructive comment, we removed age, BMI, and sex from Table 1.

Figure 1 has to be fixed correctly.

Response

We revised the presentation of Figure 1.

Table 3 is too messy, authors should consider to make it clearer.

Response

Results about subgroup analysis were removed.

Table S2 could be divided in two sections. One concerning the protein supplementation, one concerning the protein diet. As reported for line 226, authors should reconsider the included studies because they seem too different from each other.

Response

According to the reviewer’s comment, we revised the selection of studies to make a collection of more homogeneous studies. The RCTs employing the protein diet were not included in the revised manuscript. The results regarding supplementation were revised as follows:

Lines 217–223.

3.3. Protein Supplementation Characteristics

Protocols for PS are summarized in Table 1. Details of protocols for PS including additional PS in Supplementary Table S2. The protocol for protein nutrient intervention varied widely across the included RCTs. Among the 6 included RCTs, 4 prescribed essential amino acids such as leucine [31, 33, 34, 62] and 2 used branched chain amino acids [32, 63] with a supplement dose of 3.0-6.0 g/session or 8.0-40.0 g/d. In particular, PS interventions were mostly employed by 5 RCTs for patients who received total joint replacement [31-34, 62], with an intervention period of 2-7 weeks after surgery (Table S2).”

Table S2 was also revised accordingly.

Table S3 There is a bit of confusion among the included studies. I mean, almost the whole paper talks about exercise training where the patient follows a training session of several weeks. However, among the included studies there are some of them about physiotherapy rehabilitation while some others about a training session like strength training. The first ones require the therapist's guidance, the second ones can be performed independently by the patient. Authors should consider to review the selection of studies in order to make a collection of more homogeneous studies, as reported for line 236.

Response

According to the reviewer’s comment, we revised the selection of studies to make a collection of more homogeneous studies. The results regarding exercise training protocols were revised as follows:

Lines 225–231.

3.4. Exercise Training Protocol

A summary of protocols for ET is presented in Table 1, and details of the ET protocol are presented in Supplementary Table S3. Regarding the training mode of exercise, 1 RCT used preoperative home-based RET for older women with OA who were scheduled for primary unilateral total hip arthroplasty [63], and the other 5 employed postoperative physical therapy incorporating with MET for those who underwent TKR or THR.”

Table S3 was also revised accordingly.

Reviewer 2 Report

Liao et al. conducted a systematic review and meta-analysis to examine the effects of protein supplementation (PS) and exercise therapy (ET) on muscle mass and function outcomes in elderly adults with osteoarthritis (OA) in the lower extremity. The novelty in this study is in filling the current deficit in knowledge pertaining to specifically elderly populations that have OA or who underwent total joint replacement procedures. The authors have published previous meta-analyses/meta-regressions in the areas of muscle mass gain in elderly adults with OA (Liao et al., Arthritis Care, 2019) and sarcopenia (Liao et al., Nutrients, 2019). The authors are now interested in muscle mass and function PS+ET in elderly populations who have OA. The authors have presented, here, their third meta-analysis, now with using PS+ET, in elderly populations. They are addressing a critical area in elderly care, which is muscle mass gain and function in populations with OA.

Reviewer questions:

  • Table 2: I do not understand the relevance of this data, here, in the main text. You found 3 RCT studies with a medium PEDro score out of a total of your 15 final studies chosen. Hu 2019 is given the greatest weight in this analysis and they use RET. Jiang 2013 uses MET rather than RET. You then combine these three studies to gain a OR. The overall meaning of this Table, to me, is questionable. If you would like to use this data, I would move it to the Supplementary section. Could you explain more your rationale for 1.) doing this sub-analysis here in Table 2 and 2.) why you have put it in the main text? I would recommend that you move this data to the Supplementary section or omit it totally.
  • Table 3: Could you explain more about why you have chosen only 5 out of your 15 RCT studies to run an analysis on muscle mass SMD? Also, you have only 1 study for a PEDro score of equal to or above 7 (Ikeda et al., Asia Pac J Clin Nut, 2019). You have found a P value of 0.02. How did you do your statistics with only one study subgroup? Why did you not use the other 2 studies that had a PEDro score of equal to or above 7 for this analysis? I would suggest that you change the “MQ level” category to better fit the data that you have. I would like to see the SMD data for muscle mass from all 15 studies combined that have analyzed PS+ET (from Table 2). My argumentation regarding MQ level can be extended to the other categories analyzed, such as pain and muscle strength. How did you get SMD and P values with limited data? The answers to these questions should be also included early and more elaboratively in the Discussion section besides the brief mention on lines 514 - 517.
  • Figure 4. This should be moved to the Supplementary section, as it is present to indicate any bias present and does not contribute significantly to supporting your hypothesis.
  • Figures S1-S6. These are meaningful figures. Why did you put them outside of the main text and into the Supplementary section? These Figures allow the reader to easily understand the effects of PS + ET on muscle mass and function. I think that this data is relevant for the main text and would greatly improve your clinical message. In the next version, I would like to see these in the main text with appropriate description in the text. The inclusion of Table 2 and Figure 4 in the main text, to me, is not logical. Also, again, please address why you have at most 5 studies in your SMD analysis for each study/subgroup subtotal value. The interpretation of this data is greatly limited due to the scarcity in number of RCTs. The answers to these questions should be also included early and more elaboratively in the Discussion section besides the brief mention on lines 514 - 517.

Minor comments:

  • Line 28: Grammar: Parentheses should be outside and brackets should be inside.
  • Line 34: Grammar: There is a “1” before the period.
  • Table S1: Table: Please have your English Language Reviewer recheck this document. There are unnecessary parentheses and I would rather use “or” than “OR” to avoid confusion with Odds Ratio.

Author Response

Reviewer 2

Comments and Suggestions for Authors

Overall considerations

Liao et al. conducted a systematic review and meta-analysis to examine the effects of protein supplementation (PS) and exercise therapy (ET) on muscle mass and function outcomes in elderly adults with osteoarthritis (OA) in the lower extremity. The novelty in this study is in filling the current deficit in knowledge pertaining to specifically elderly populations that have OA or who underwent total joint replacement procedures. The authors have published previous meta-analyses/meta-regressions in the areas of muscle mass gain in elderly adults with OA (Liao et al., Arthritis Care, 2019) and sarcopenia (Liao et al., Nutrients, 2019). The authors are now interested in muscle mass and function PS+ET in elderly populations who have OA. The authors have presented, here, their third meta-analysis, now with using PS+ET, in elderly populations. They are addressing a critical area in elderly care, which is muscle mass gain and function in populations with OA.

Response

We thank all the reviewers for their comprehensive review and their comments regarding our manuscript. We have made all necessary modifications to our originally submitted manuscript (nutrients-882515), based on reviewers’ comments, point by point.

In addition, our revised manuscript has been edited by a professional editing company (Wallace Academic Editing).

Reviewer questions:

  • Table 2: I do not understand the relevance of this data, here, in the main text. You found 3 RCT studies with a medium PEDro score out of a total of your 15 final studies chosen. Hu 2019 is given the greatest weight in this analysis and they use RET. Jiang 2013 uses MET rather than RET. You then combine these three studies to gain a OR. The overall meaning of this Table, to me, is questionable. If you would like to use this data, I would move it to the Supplementary section. Could you explain more your rationale for 1.) doing this sub-analysis here in Table 2 and 2.) why you have put it in the main text? I would recommend that you move this data to the Supplementary section or omit it totally.

Response  

In accordance with the reviewer’s comment, we move the original Table 2 to the Supplementary section, namely Table S4.

  • Table 3: Could you explain more about why you have chosen only 5 out of your 15 RCT studies to run an analysis on muscle mass SMD? Also, you have only 1 study for a PEDro score of equal to or above 7 (Ikeda et al., Asia Pac J Clin Nut, 2019). You have found a P value of 0.02. How did you do your statistics with only one study subgroup? Why did you not use the other 2 studies that had a PEDro score of equal to or above 7 for this analysis? I would suggest that you change the “MQ level” category to better fit the data that you have. I would like to see the SMD data for muscle mass from all 15 studies combined that have analyzed PS+ET (from Table 2). My argumentation regarding MQ level can be extended to the other categories analyzed, such as pain and muscle strength. How did you get SMD and P values with limited data? The answers to these questions should be also included early and more elaboratively in the Discussion section besides the brief mention on lines 514 - 517.

Response

Following the comments of reviewer 1, we revised the study collection and selection; and finally, 6 RCTs were included in the revised manuscript. Among the newly included RCTs, only 4 RCTs had muscle mass measures (please refer to Table 1) which were used for meta-analysis. Accordingly, we reperformed all analyses for each primary and secondary outcome. The subgroup analysis was not performed due to that no significant heterogeneity was identified in all outcome measures. Therefore, we removed the statements about results and discussions of subgroup analyses throughout the revised manuscript. We made statements to clarify it in Discussion section as follows:

Lines 361–363.  

“The subgroup analyses were not performed due to that no significant heterogeneity was identified in all outcome measures.”

  • Figure 4. This should be moved to the Supplementary section, as it is present to indicate any bias present and does not contribute significantly to supporting your hypothesis.

Response  

Following the reviewer’s comment, we move the original Figure 4 to the Supplementary section, namely Figure S8.

  • Figures S1-S6. These are meaningful figures. Why did you put them outside of the main text and into the Supplementary section? These Figures allow the reader to easily understand the effects of PS + ET on muscle mass and function. I think that this data is relevant for the main text and would greatly improve your clinical message. In the next version, I would like to see these in the main text with appropriate description in the text. The inclusion of Table 2 and Figure 4 in the main text, to me, is not logical. Also, again, please address why you have at most 5 studies in your SMD analysis for each study/subgroup subtotal value. The interpretation of this data is greatly limited due to the scarcity in number of RCTs. The answers to these questions should be also included early and more elaboratively in the Discussion section besides the brief mention on lines 514 - 517.

Response

Following the reviewer’s comment, we move the original Figures S1–S6 to the main text, namely Figures 2–5.

Minor comments:

  • Line 28: Grammar: Parentheses should be outside and brackets should be inside.

Response

Lines 30–31.

We revised the statement as follows:“The PS+ET group exhibited significant improvements in the muscle mass (standard mean difference [SMD] = 1.13; P < 0.00001), ……” 

  • Line 34: Grammar: There is a “1” before the period.

Response

The word “1” was removed.

  •  
  • Table S1: Table: Please have your English Language Reviewer recheck this document. There are unnecessary parentheses and I would rather use “or” than “OR” to avoid confusion with Odds Ratio.

Response

The revised manuscript (including Table S1) has been edited by a professional editing company (Wallace Academic Editing).

Round 2

Reviewer 2 Report

Thank you for your significant improvements to your manuscript. After these revisions, I have highly recommended for it to be published in Nutrients. I am confident that this review will provide the light for future studies in this important area of research.

Some minor comments that require you to edit regarding the language:

Line 70: “remain inconsistent.”

Line 72: “conflicting”

Line 72: “based on the fact”

Line 267: Remove “Flowchart of enrollment of the studies.”